# Deliberation in Latent Space via Differentiable Cache Augmentation

Luyang Liu [1]  Jonas Pfeiffer [1]  Jiaxing Wu [1]  Jun Xie [1]  Arthur Szlam [1]

## Abstract

Techniques enabling large language models (LLMs) to "think more" by generating and attending to intermediate reasoning steps have shown promise in solving complex problems. However, the standard approaches generate sequences of discrete tokens immediately before responding, and so they can incur significant latency costs and be challenging to optimize. In this work, we demonstrate that a frozen LLM can be augmented with an offline coprocessor that operates on the model's key-value (kv) cache. This coprocessor augments the cache with a set of latent embeddings designed to improve the fidelity of subsequent decoding. We train this coprocessor using the language modeling loss from the decoder on standard pretraining data, while keeping the decoder itself frozen. This approach enables the model to learn, in an end-to-end differentiable fashion, how to distill additional computation into its kv-cache. Because the decoder remains unchanged, the coprocessor can operate offline and asynchronously, and the language model can function normally if the coprocessor is unavailable or if a given cache is deemed not to require extra computation. We show experimentally that when a cache is augmented, the decoder achieves lower perplexity on numerous subsequent tokens. Furthermore, even without any task-specific training, our experiments demonstrate that cache augmentation consistently improves performance across a range of reasoning-intensive tasks.

## 1. Introduction

Recent research (Wei et al., 2022; Kojima et al., 2022; Wu et al., 2024) has shown that enabling large language models (LLMs) to generate, or even search over, intermediate sequences of steps before producing a final answer can significantly improve performance on reasoning tasks. More broadly, providing LLMs with the ability to allocate compute adaptively during generation can lead to more effective generation within a fixed compute budget (Schuster et al., 2022). However, at a high level, many of these "extra thinking" approaches are similar in that their sequences of intermediate outputs are discrete, making them difficult to train in an end-to-end fashion, and in that their extra "thinking" (i.e. computation) is performed just-in-time, as part of the output generating process.

In this work, we introduce a fundamentally different approach, inspired by the literature on kv-cache compression (Mu et al., 2024; Ge et al., 2024). Our approach takes a step towards LLMs that can deliberate on their memories (encoded in the kv-cache), and distill these deliberations into a form usable for subsequent tasks. Specifically, our method processes the transformer's cache and augments it with a set of soft tokens produced in a single forward pass–not sequentially. This extra processing is performed by a separate model, which we refer to as a "coprocessor", while the base transformer remains frozen. Once the kv-cache is augmented with the coprocessor's output (which we term "latent embeddings"), decoding proceeds as normal until the coprocessor is called again. This approach offers the following key advantages:

**End-to-end Differentiability**: Our framework enables end-to-end backpropagation during coprocessor training, facilitating efficient optimization without the need for reinforcement learning techniques. We leverage the standard language-modeling loss on pre-training data, making the method scalable.

**Asynchronous Operation**: Because cache augmentation improves results many tokens beyond the augmentation point, and because the base transformer remains frozen during coprocessor training, asynchronous coprocessor operation becomes feasible. This contrasts with existing methods where additional computation occurs sequentially, and online. Our approach opens the door to models that can strategically bank computation by deliberating and refining their internal memory, independent of composing a response to a query.

We evaluate our method using Gemma-2 (Team-Gemma et al., 2024) models pretrained on a diverse dataset mixture.

[1]Google DeepMind. Correspondence to: Luyang Liu <luyang.liu@google.com>, Arthur Szlam <aszlam@google.com>.

*Proceedings of the $42^{nd}$ International Conference on Machine Learning*, Vancouver, Canada. PMLR 267, 2025. Copyright 2025 by the author(s).

Our experiments demonstrate that without any fine-tuning on specific downstream tasks, our approach consistently improves performance across a range of reasoning-intensive tasks. We observed that increasing the number of injected latent embeddings generally leads to better performance. For example, we observe a 10.05% improvement on GSM8K and a 4.70% improvement on MMLU, when augmenting the Gemma-2 2B model with 64 latent embeddings. These results highlight the potential of enhancing LLMs with kv-cache coprocessing for augmenting model capabilities.

## 2. Methodology

We enhance a frozen LLM by training a coprocessor that inputs a key-value (kv) cache, and augments it with a set of soft tokens. This section details the architecture and training process of our approach.

### 2.1. Problem statement

Given an input $x$ and a desired target output $y$, and a pretrained, frozen LLM parameterized by $\theta$, we seek to learn a coprocessor, denoted by $f$. This coprocessor takes the kv-cache $(k_{\theta,x}, v_{\theta,x})$ generated by the frozen LLM when processing the input $x$ as input, and outputs a sequence of latent representations $z$:

$$f(k_{\theta,x}, v_{\theta,x}) \longrightarrow z \qquad (1)$$

The objective of learning $f$ is to produce latent embeddings $z$ that, when combined with the input $x$, improve the frozen LLM's ability to generate the correct target $y$. Specifically, we aim to maximize the expected log-likelihood of the target $y$ given the input $x$ and the learned latent embeddings $z$, as predicted by the frozen LLM:

$$\max E_x[\log p_\theta(y|x, z)] \qquad (2)$$

### 2.2. Model architecture

Our proposed architecture enhances a frozen, pretrained LLM with a dedicated coprocessor module operating on the kv-cache. As illustrated in Figure 1, the interaction between these components unfolds in three stages:

**KV-cache Generation:** The input sequence, $x$, is first processed by the frozen LLM to generate its corresponding kv-cache $(k_{\theta,x}, v_{\theta,x})$. This cache encapsulates the LLM's internal representations of the input. Crucially, the LLM's weights remain frozen throughout the entire process.

**Augmentation:** The kv-cache is then passed to the coprocessor module, which adopts the same model architecture as the pretrained LLM and therefore has an identical number of parameters as the frozen LLM. The coprocessor also receives a sequence of distinct extra soft tokens with trainable embeddings. These tokens do not correspond to actual

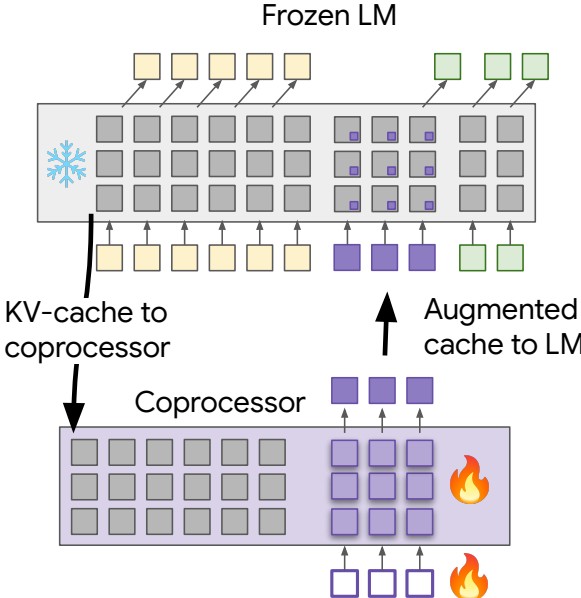

*Figure 1.* Overview of the proposed architecture. The input sequence is processed by a frozen LLM, generating a kv-cache. This cache is then passed to a coprocessor, along with trainable soft tokens. The coprocessor outputs latent embeddings which are used to augment the original kv-cache before being fed back into the LLM for output generation.

words or sub-words but serve as abstract prompts for the coprocessor. The coprocessor ingests the kv-cache and these tokens to produce a sequence of latent embeddings, $z$.

**LLM Generation with Augmented Context:** Finally, $z$ is appended to the original kv-cache. This augmented cache is then fed back into the frozen LLM, providing it with enriched contextual information derived from the coprocessor. The LLM then proceeds to generate the output sequence, $y$, conditioned on both the original input $x$ and the coprocessor's output $z$. This allows the LLM to leverage the coprocessor's latent inferences without requiring it to explicitly verbalize intermediate steps.

Training focuses solely on optimizing the coprocessor and trainable embeddings' weights. The coprocessor shares the same model architecture as the pretrained LLM, and its weights are initialized with the pretrained weights of the LLM. The loss is calculated on the final output $y$, and backpropagation is used to update only the coprocessor's parameters. This targeted training approach allows for efficient fine-tuning without altering the pretrained LLM. In practice, the coprocessor's augmentation can potentially be performed offline and asynchronously, in parallel with the LLM's decoding process. This could enable continuous refinement of the LLM's contextual memory, leading to improved efficiency and faster response times.

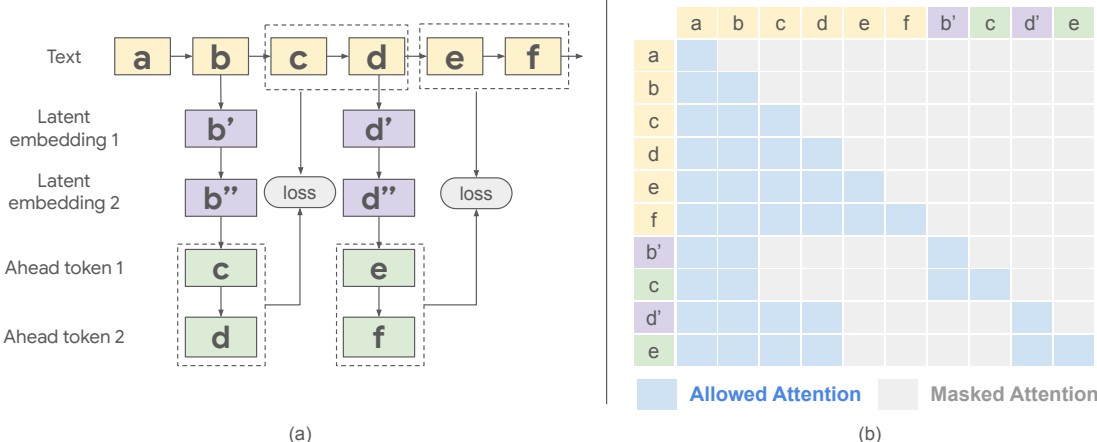

*Figure 2.* Our coprocessor training framework. (a) Illustration of multi-position augmentation and ahead token prediction. For each selected augmentation position, latent embeddings are generated by the coprocessor and inserted after the corresponding token's embedding. The target tokens for prediction ("ahead tokens") are then appended. A causal mask is applied to all sequences following these insertion points. (b) Structure of the modified input and attention mask for model training. We show an example of 1 latent embedding and 1 ahead token here for simplicity.

## 2.3. Pretraining setup

We employ a pretraining strategy designed to encourage the coprocessor to learn augmentations that will be useful for predicting larger segments of text beyond the next token after the augmentation. This strategy involves generating augmentations for multiple randomly selected positions and training the coprocessor to predict many tokens ahead.

Instead of training on a single split of a sequence into input $x$ and target $y$ (which limits scalability), we augment at multiple points within each sequence. As shown in Figure 2 (a), given an input text sequence (e.g., "a b c d e f"), we randomly select a subset of positions (e.g., "b" and "d"). For each selected position, the coprocessor generates a configurable number of latent embeddings (e.g., b', b" and d', d" in the figure, where the number of latent embeddings is a hyperparameter $N_L$) in one transformer forward call. The training objective is to predict a number of tokens (another hyperparameter $N_A$) beyond the placement of the augmentation, in a teacher-forcing way. For instance, if "b" is chosen, and we are predicting two tokens ahead, the coprocessor uses the generated latent embeddings (b', b") and the kv-cache of the preceding text "a" and "b" to predict "c" and "d". This process can be viewed as a form of latent space interpolation: the coprocessor learns to bridge the gap between the known preceding context ("a", "b") and the future context ("c", "d") by generating meaningful latent representations (b', b"). Similarly, if "d" is chosen, the targets would be "e" and "f", based on d', d" and the preceding context "a b c d". This approach is similar to the parallel decoding introduced in Quiet-Star (Zelikman et al., 2024), but since we generate latent embeddings instead of thoughts in token space, our approach has the advantages of fully differentia-

bility while keeping the LLM frozen. Furthermore, because the base LLM remains frozen, the coprocessor can be called asynchronously and its computations can be performed in parallel with the LLM's decoding operation, and there is no need to insert between every pair of consecutive tokens.

We implement an efficient training framework by modifying the input, attention mask, position index, and target, to enable training everything together in one forward pass. Figure 2(b) illustrates how we modify the input and attention mask to enable training on multiple positions. Instead of sequentially processing each augmentation position in multiple decoder forward calls, we construct a single, extended input sequence and corresponding attention mask. This allows us to compute the loss from all selected augmentation points in parallel. For each selected augmentation position, we insert the generated latent embeddings after the corresponding token's embedding in the input sequence. The ahead tokens for that position are then appended after the latent embeddings. This creates a concatenated input sequence containing the original text followed by multiple groups of latent embeddings and their corresponding ahead tokens. The attention mask is constructed to ensure correct causal attention. The original tokens attend to all preceding original tokens, as usual. Crucially, the latent embeddings and their corresponding ahead tokens only attend to the original tokens preceding their insertion point. They are masked from attending to any subsequent original tokens, other latent embeddings, or other ahead tokens as shown in Figure 2(b). The resulting output is then used to calculate the loss and train the coprocessor.

Rather than capturing different aspects of a single token's context, these embeddings are trained to generate informa-

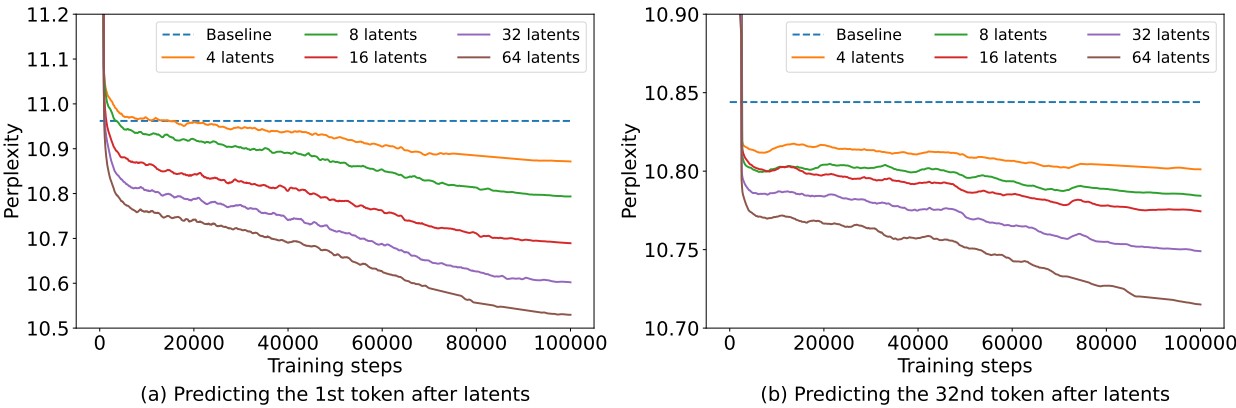

*Figure 3.* Validation perplexity of the baseline frozen Gemma-2 2B model and augmented models with varying numbers of latents (8, 16, 32, 64), when predicting the 1st and 32nd tokens following latent augmentation. Lower perplexity indicates better performance.

tion useful for predicting future tokens, effectively enabling the model to perform a form of "latent thinking" before making predictions. This contrasts with standard next-token prediction and allows the coprocessor to learn more generalizable representations. By learning to anticipate future tokens in this manner, the coprocessor develops a stronger understanding of sequential dependencies within text, which proves valuable in downstream tasks.

## 3. Experiments

We validate our approach using the frozen Gemma-2 2B model. Our augmented Gemma-2 models, with only the coprocessor being trained and the decoder-only LLM kept frozen, are trained on the same 2 trillion token, primarily-English dataset used for Gemma-2 pretraining (Team-Gemma et al., 2024), following the setup described in Section 2.2. This dataset includes a variety of sources, such as web documents, code, and scientific articles. We trained the model for 100,000 steps using a batch size of 1024, packed sequences of length 2048, 16 ahead tokens ($N_A$), and 128 randomly sampled augmentation positions ("traces") for all training experiments. Importantly, no task-specific training is performed for any of the experiments; all training is done on the pretraining dataset.

The frozen Gemma-2 model serves as our primary baseline. We also experimentally verified that continued training of the base Gemma-2 model on this same pretraining data subset results in negligible differences in perplexity or downstream performance, given its comprehensive initial pretraining on this data. Consequently, this was not pursued as a formal baseline for all reported comparisons.

### 3.1. Perplexity Evaluation

Our augmented Gemma model is able to achieve lower perplexity on the validation dataset compared to the pretrained Gemma model on many tokens ahead, even beyond the ahead token $N_A$ we defined during training. We evaluate this using a proprietary validation dataset (Same as the one used in Gemma (Team-Gemma et al., 2024)) and evaluate the effect of augmenting the frozen Gemma-2 2B LLM's kv-cache on future token prediction. For each sequence in the validation set, we generate $N_L$ latent embeddings after each token using our coprocessor. These embeddings are then used to augment the cache at each token position. We then measure the model's ability to predict the $n$-th future token. Specifically, the "1st token" perplexity measures the model's performance predicting the token immediately following the inserted latent embeddings. The "32nd token" perplexity measures the model's performance predicting the token 32 positions ahead, given the context preceding the latent embeddings, the embeddings themselves, and the following 31 tokens. This demonstrates that even though we train with $N_A = 16$, the benefits of cache augmentation extend beyond this range, improving predictions even at position 32. Figure 3 presents perplexity curves during training for the baseline frozen Gemma-2 2B model and our augmented models using $N_L$=8, 16, 32, and 64 latent embeddings. Across all latent sizes, our approach consistently reduces perplexity, with the improvement scaling with the number of latent embeddings. This demonstrates that augmenting the cache with the coprocessor improves both short-range and longer-range prediction accuracy.

Table 1 quantifies the perplexity reduction achieved by our cache augmentation approach. The consistent reduction, generally correlating with the number of latents, confirms that the benefit of our method extends to multiple subsequent token predictions, suggesting improved internal represen-

| Position | 8 Latents | 16 Latents | 32 Latents | 64 Latents |
|---|---|---|---|---|
| 1 | -1.53% | -2.48% | -3.28% | -3.94% |
| 2 | -1.67% | -2.41% | -3.15% | -3.70% |
| 4 | -1.39% | -1.98% | -2.66% | -3.17% |
| 8 | -1.22% | -1.56% | -2.11% | -2.61% |
| 16 | -0.85% | -1.08% | -1.50% | -1.88% |
| 32 | -0.55% | -0.64% | -0.88% | -1.20% |

*Table 1.* Relative perplexity reduction (in %) achieved by augmented models compared to the baseline, for various numbers of latent embeddings and positions following latent augmentation. "Position" indicates the token position relative to the augmentation point. (e.g., Position 1 is the immediately following token).

tations within the decoder, leading to more accurate and coherent generation.

### 3.2. Public Benchmark Evaluation

We evaluated cache augmentation on a range of public benchmarks spanning natural language understanding and reasoning tasks (Table 2). In this setting, we only call the coprocessor *once*, at the end of the prompt. Our method consistently improves performance compared to the baseline frozen Gemma-2 2B model, with particularly substantial gains on reasoning-intensive benchmarks. Several tasks, including MMLU, GSM8K, TriviaQA, NQ, and MATH, exhibit a strong correlation between the number of latent embeddings and performance improvement. For example, on GSM8K, accuracy steadily climbs from a +1.29% gain with 4 latent embeddings to a notable +10.05% with 64. Similarly, MATH improves from -0.12% with 4 to +2.06% with 64, and MMLU shows a jump from +0.45% with 4 to +4.70% with 64. This trend suggests that for certain challenging reasoning tasks, providing more latent embeddings allows the model to perform more extensive "thinking" in the latent space, significantly enhancing its reasoning capabilities.

Other reasoning tasks, including ARC-e/c, Winogrande, and Boolq, also show improvements with increasing latent counts. While some tasks, such as AGIEval, BBH, and HumanEval, show less pronounced improvements or occasional performance dips with higher latent embedding counts, our method still frequently provides a benefit. This broad improvement across diverse benchmarks underscores the effectiveness and general applicability of cache augmentation for enhancing frozen language models.

We further provide analysis in the appendix, showing that our method's performance scales with increasing training data (Section A.1) and that it effectively adapts to downstream tasks (Section A.2).

### 3.3. Comparison with other baselines and variations

#### 3.3.1. PAUSE TOKEN

We compare our approach with a closely related baseline: the Pause Token method (Goyal et al., 2023). Pause Token introduces trainable embeddings inserted between the input ($x$) and output ($y$) sequences, encouraging the LLM to perform latent "thinking" before generating the output. The crucial distinction between our approach and Pause Token lies in how these latent embeddings are generated. While Pause Token utilizes fixed and pretrained embeddings that do not condition on the input $x$, our method employs a coprocessor that generates context-dependent, dynamic embeddings based on the input. This allows our approach to tailor the latent representations to the specific input, potentially leading to more effective reasoning.

Table 3 directly compares the performance of the baseline Gemma-2 2B model against both Pause Token and our approach, with the latter two using 32 embeddings. Notably, the Pause Token model was trained using the same training data and under the same experimental setup as our method. On the validation set, our method achieves a perplexity of 10.60 on the first token prediction, significantly lower than both the baseline (10.96) and Pause Token (11.63). Furthermore, our method achieves an accuracy of 26.76% on the GSM8K dataset, outperforming both the baseline (21.38%) and Pause Token (22.37%). These improvements underscore the effectiveness of our dynamic, contextually-informed embeddings, which provide a richer representation compared to the fixed embeddings in Pause Token, leading to better next token prediction and improved performance on reasoning tasks.

#### 3.3.2. ZERO-SHOT COT

Our technique can be viewed as a form of latent Chain-of-Thought (CoT) prompting. Therefore, we compare our approach to standard zero-shot CoT (Kojima et al., 2022), which involves appending "Let's think step by step" to the input prompt. While zero-shot CoT can be effective, it relies on the LLM to generate intermediate reasoning steps token by token, which can be computationally expensive during inference. Our method, on the other hand, generates latent embeddings in a single forward pass, potentially offering a more efficient approach to guiding reasoning.

Table 4 presents the accuracy on GSM8K for the baseline Gemma-2 2B model, zero-shot CoT, and our approach with 16 and 32 latent embeddings. Our method shows clear improvements. With 16 latent embeddings, we achieve an accuracy of 24.72%, surpassing both the baseline (21.38%) and zero-shot CoT (23.20%). Performance further improves to 26.76% with 32 embeddings. This suggests that our learned, context-dependent latent embeddings provide a

| Benchmark | Metric | Baseline | 4 Latents | 8 Latents | 16 Latents | 32 Latents | 64 Latents |
|-----------|--------|----------|-----------|-----------|------------|------------|------------|
| MMLU | 5-shot | 52.00 | 52.45 (+0.45) | 52.24 (+0.24) | 52.34 (+0.34) | 54.61 (+2.61) | **56.70 (+4.70)** |
| GSM8K | 8-shot | 21.38 | 22.67 (+1.29) | 23.12 (+1.74) | 24.72 (+3.34) | 26.76 (+5.38) | **31.43 (+10.05)** |
| DROP | 3-shot, F1 | 53.69 | 54.64 (+0.95) | 54.91 (+1.23) | 56.23 (+2.55) | 57.37 (+3.68) | **57.77 (+4.08)** |
| ARC-e | 0-shot | 80.56 | 81.52 (+0.97) | 81.57 (+1.01) | 83.12 (+2.57) | 83.04 (+2.48) | **83.67 (+3.11)** |
| ARC-c | 0-shot | 50.26 | 51.28 (+1.02) | 52.39 (+2.13) | 53.24 (+2.99) | **54.44 (+4.18)** | **54.44 (+4.18)** |
| MATH | 4-shot | 16.50 | 16.38 (-0.12) | 16.78 (+0.28) | 17.00 (+0.50) | 17.18 (+0.68) | **18.56 (+2.06)** |
| Winogrande | 0-shot | 64.01 | 65.35 (+1.34) | 65.35 (+1.34) | 66.30 (+2.29) | 66.30 (+2.29) | **66.61 (+2.60)** |
| PIQA | 0-shot | 78.18 | 78.62 (+0.44) | 78.67 (+0.49) | 78.94 (+0.76) | 78.94 (+0.76) | **79.00 (+0.82)** |
| SIQA | 0-shot | 51.79 | 51.59 (-0.20) | 51.64 (-0.15) | 51.74 (-0.05) | **52.30 (+0.51)** | 52.00 (+0.20) |
| HellaSwag | 0-shot | 73.77 | 74.41 (+0.64) | 74.41 (+0.64) | 74.82 (+1.05) | 75.04 (+1.27) | **75.31 (+1.54)** |
| Boolq | 0-shot | 75.41 | 75.29 (-0.12) | 77.22 (+1.80) | **78.17 (+2.75)** | 77.03 (+1.62) | 76.91 (+1.50) |
| MBPP | 3-shot | 30.40 | 29.00 (-1.40) | 31.60 (+1.20) | 31.20 (+0.80) | 31.40 (+1.00) | **31.80 (+1.40)** |
| AGIEval | 3-5-shot | 31.71 | 32.18 (+0.47) | 30.04 (-1.67) | 31.32 (-0.38) | 32.78 (+1.07) | **33.85 (+2.14)** |
| TriviaQA | 5-shot | 60.29 | 60.30 (+0.01) | 60.83 (+0.54) | 61.43 (+1.14) | 62.05 (+1.76) | **62.23 (+1.94)** |
| NQ | 5-shot | 17.14 | 17.35 (+0.21) | 17.89 (+0.75) | 18.16 (+1.02) | 18.91 (+1.77) | **19.20 (+2.06)** |
| HumanEval | pass@1 | 19.51 | 18.29 (-1.22) | 19.51 (+0.00) | 20.73 (+1.22) | 20.73 (+1.22) | **22.56 (+3.05)** |
| BBH | 3-shot | 42.22 | 42.36 (+0.14) | 42.37 (+0.15) | 42.53 (+0.31) | 42.48 (+0.26) | **42.64 (+0.41)** |

*Table 2.* Performance of baseline and augmented models across various benchmarks. Results are shown for the baseline (frozen Gemma-2 2B pretrained model) and the model augmented with a learned coprocessor using 4, 8, 16, 32, and 64 latent embeddings, respectively. Results are reported for zero/few-shot settings as indicated in the "Metric" column. Results are accuracy (in %) if not specified otherwise. Improvements over the baseline are shown in parentheses. In this setting, the coprocessor is called *once*, at the end of the prompt.

| Method | Validation set perplexity (↓) | GSM8K 8-shot accuracy (↑) |
|--------|-------------------------------|---------------------------|
| Baseline Gemma 2B | 10.96 | 21.38 |
| Pause Token | 11.63 | 22.37 |
| Cache Augmentation | **10.60** | **26.76** |

*Table 3.* Comparison between the baseline Gemma-2 2B model, the Pause Token method (Goyal et al., 2023) (using 32 embeddings), and our approach (also using 32 embeddings). Lower perplexity indicates better next token prediction. Higher accuracy indicates better performance on GSM8K.

| Baseline | 0-shot CoT | 16 Latents | 32 Latents |
|----------|-----------|------------|------------|
| 21.38 | 23.20 | 24.72 | **26.76** |

*Table 4.* Accuracy on GSM8K 8-shot for the baseline Gemma-2 2B model, zero-shot Chain-of-Thought (CoT) prompting, and our approach with 16 and 32 latent embeddings.

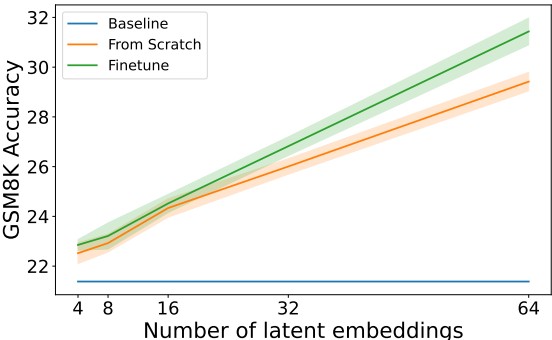

*Figure 4.* Finetuning the coprocessor from Gemma-2 2B pretrained weights significantly improves GSM8K accuracy compared to training from scratch. Lines represent the mean and shaded areas represent the 95% confidence interval, both estimated from the last 5 checkpoints.

more efficient and effective mechanism for guiding reasoning compared to the generic prompt and sequential token generation of zero-shot CoT.

It is worth noting that for some other benchmarks, such as MATH and HumanEval, the performance of zero-shot CoT was closer to the baseline Gemma-2 2B model. GSM8K was highlighted here as it demonstrated a notable improvement with our method over both the baseline and zero-shot CoT.

### 3.3.3. ALTERNATIVE COPROCESSOR CONFIGURATIONS

We explored alternative configurations for our coprocessor to assess the importance of design choices. Our default setup involves finetuning the pretrained LLM to serve as the coprocessor (i.e., the coprocessor's weights are initialized with the pretrained weights of the LLM), which serves as

the primary comparison point for the following experiments.

**Training the Coprocessor from scratch:** We also investigated training the coprocessor from scratch–randomly initializing its weights rather than finetuning from the pretrained weights of Gemma-2 2B. While training from scratch improves performance on all downstream tasks compared to the baseline, finetuning from pretrained weights yields even better results. This suggests that the coprocessor benefits from the foundational knowledge encoded in the pretrained LLM. Figure 4 illustrates this improvement in GSM8K accuracy as the number of latent embeddings increases, with the finetuned model consistently outperforming the model trained from scratch. Results of other benchmarks can be found in Table 7 in the Appendix Section A.3.

**LoRA Finetuning the pretrained LLM as the Coprocessor:** In addition to full finetuning the pretrained LLM and from-scratch training, we explored the efficacy of Low-Rank Adaptation (LoRA) (Hu et al., 2021) for tuning the coprocessor from the pretrained LLM's weights. LoRA freezes the pretrained model weights and introduces trainable rank-decomposition matrices, significantly reducing the number of trainable parameters. For instance, LoRA with a rank of 64 for the Gemma-2 2B model adds approximately 2% extra trainable parameters compared to the base LLM. This approach offers substantial memory benefits, as only the relatively small LoRA weights need to be stored in addition to the base model. We experimented with LoRA using ranks of 64 and 128, comparing their performance on GSM8K to the baseline Gemma-2 2B model, the from-scratch training approach discussed above, and our fully finetuned coprocessor. As shown in Table 5, which presents results using 32 latent embeddings for all methods, LoRA finetuning achieves reasonable improvements over the baseline, demonstrating that even a parameter-efficient approach can effectively train the coprocessor for improved reasoning. Specifically, LoRA with rank 64 achieved an accuracy of 23.35%, while LoRA with rank 128 reached 24.03%. These results fall between the baseline performance (21.38%) and the performance achieved by from-scratch training (25.78%), indicating that while the LoRA-tuned coprocessor benefits from the pretrained weights, generating high-quality latent embeddings for effective reasoning appears to require more substantial parameter updates than those provided by parameter-efficient methods like LoRA. While these results are not as strong as full finetuning (26.76%), they represent a notable improvement over the baseline and highlight the potential of LoRA for efficient training/inference of our coprocessor, especially in memory-constrained environments.

**Augmentation using Last Layer's Activations:** Instead of using the kv-cache as input to the coprocessor, we experimented with providing the last layer's activations from the frozen LLM, concatenated with the soft token embeddings. This approach, using 32 latent embeddings, yielded a perplexity of 10.81 on the validation set and an accuracy of 23.20% on the GSM8K benchmark under the same training setup. Both metrics are notably worse than those achieved with kv-cache augmentation, which resulted in a perplexity of 10.69 and a GSM8K accuracy of 26.76% (also with 32 latent embeddings). We hypothesize that the last layer's activations alone do not provide as rich a representation for the coprocessor as the information aggregated across multiple layers in the kv-cache, hindering both next-token prediction (reflected in the higher perplexity) and reasoning ability (reflected in the lower GSM8K accuracy).

| Method | GSM8K Accuracy |
|---|---|
| Baseline | 21.38 |
| LoRA (Rank 64) | 23.35 |
| LoRA (Rank 128) | 24.03 |
| From Scratch Training | 25.78 |
| Full Finetuning | **26.76** |

*Table 5.* GSM8K accuracy comparison of different finetuning methods for the coprocessor, all using 32 latent embeddings. LoRA offers a memory-efficient alternative to full finetuning, achieving reasonable performance gains.

### 3.4. Impact of the number of ahead token in training

We investigated the impact of varying the number of ahead tokens—the number of future tokens the model is trained to predict—during coprocessor training. While larger lookahead improves perplexity on later tokens, it often leads to higher perplexity on earlier tokens. Though learning rate scaling might mitigate this, we empirically chose 16 ahead tokens for most experiments in this paper, given its strong performance on GSM8K, as shown in the Table 6.

## 4. Related Work

### 4.1. Chain-of-Thought Reasoning in LLMs

Limitations in eliciting complex reasoning from LLMs through standard prompting have motivated research into prompting strategies that encourage intermediate reasoning steps. Chain-of-Thought (CoT) prompting (Wei et al., 2022) significantly improved reasoning performance by prompting LLMs to "think step by step". Subsequent work explored zero-shot CoT (Kojima et al., 2022; Zhou et al., 2023), aggregating multiple reasoning paths (Wang et al., 2022), internalizing the intermediate reasoning steps (Deng et al., 2024), verifying generation steps (Lightman et al., 2023), and broader search spaces for reasoning trajectories, such as Tree-of-Thought (Yao et al., 2024; Wang and Zhou, 2024). Other approaches leverage reinforcement learning to optimize the reasoning process based on final answer accuracy or target text likelihood (e.g., StaR (Zelikman et al., 2022), TRICE (Hoffman et al., 2024), Quiet-STaR (Zelikman et al., 2024)). While effective, these methods are often constrained by the expressiveness of natural language and can be computationally expensive due to the sequential generation of reasoning steps, both during training and inference. On the other hand, our differentiable cache augmentation approach could be viewed as complementary to these RL techniques. For instance, RL could potentially be used to further optimize the generation of latent embeddings by the coprocessor, or our method could provide an improved base model upon which RL-based reasoning strategies are applied.

| Baseline | 4 Ahead | 8 Ahead | 16 Ahead | 32 Ahead |
|----------|---------|---------|----------|----------|
| 21.38 | 24.03 (+2.65) | 24.11 (+2.73) | **24.72** (+3.34) | 23.73 (+2.35) |

*Table 6.* GSM8K accuracy for varying numbers of ahead tokens during coprocessor training. 16 ahead tokens achieves the highest accuracy (24.72%, +3.34% over the baseline of 21.38%). 16 latent embeddings are used for all these experiments.

### 4.2. Latent Space Reasoning

Previous research has investigated the role of latent transformer computations in LLMs' reasoning abilities. (Cheng and Durme, 2024) elicits reasoning with a short sequence of continuous embeddings, aiming for greater inference efficiency. As demonstrated in (Biran et al., 2024), a sequential latent reasoning pathway in LLMs is identified for multi-hop reasoning problems. (Shalev et al., 2024) revealed that the middle layers of LLMs produce highly interpretable embeddings, representing a set of potential intermediate answers for multi-hop queries. To improve LLMs' latent reasoning ability, researchers have proposed to augment LLMs with meta tokens. The Pause Token method (Goyal et al., 2023), closely related to our work, introduces trainable embeddings inserted between input and output sequences to encourage latent "thinking". Similarly, a recent work (Pfau et al., 2024) also studied the circumstances under which causal transformers are able to learn to utilize intermediate dummy tokens (e.g. the filler token used in this work). Unlike these studies which employ pretrained embeddings, our work generates latent tokens dynamically based on the input. More recently, COCONUT (Hao et al., 2024) introduced a new reasoning paradigm by utilizing LLMs' hidden states as input embeddings in latent space. While both our approach and COCONUT explore reasoning in latent space, key distinctions exist. Our method maintains a strictly frozen base LLM and generates multiple latent embeddings in parallel in a single forward pass of the coprocessor, trained on general pretraining data. In contrast, COCONUT focuses on generating sequential continuous CoT like embeddings and often involves task-specific fine-tuning and multi-stage training to internalize reasoning, without the constraint of a frozen base model.

### 4.3. KV-Cache Compression

KV-cache compression is a technique used to reduce the size of the transformer's kv-cache, the memory storing past activations, for efficient storage and faster computation. (Ge et al., 2024) propose an in-context autoencoder (ICAE) to compress the context into concise embeddings for the kv-cache. (Mu et al., 2024) introduce the concept of "gist tokens" to learn compressed representations of the kv-cache. Alternatively, prior work (Li et al., 2023) also improves the efficiency of LLMs by identifying and removing redundant information from the input, making it more compact and easier to process.

While our approach also leverages the kv-cache, our motivation is fundamentally different. Rather than focusing on compression, we aim to augment the kv-cache with latent embeddings produced by an offline coprocessor, thereby enhancing the transformer's reasoning capabilities without modifying its architecture. This allows us to improve the fidelity of further decoding and boost performance on reasoning-intensive tasks. Our work is inspired by the idea of deliberation, where the coprocessor can "think" in the latent space by processing the kv-cache and generating meaningful embeddings that guide the transformer's subsequent generation.

### 4.4. Augmenting LLMs with External Modules

Extensive research has focused on augmenting pretrained LLMs with external modules for improved efficiency and performance (Pfeiffer et al., 2023). Parameter-efficient fine-tuning methods like prompt tuning (Lester et al., 2021), prefix tuning (Li and Liang, 2021), and adapters have also been explored. Adapters, first introduced for computer vision by Rebuffi et al. (2017; 2018) and later popularized in NLP by Houlsby et al. (2019), insert small, trainable modules into the LLM. LoRA (Hu et al., 2021) further improves adapter efficiency by decomposing weight updates into low-rank matrices. Furthermore, multimodal models like Flamingo (Alayrac et al., 2022), CoCa (Yu et al., 2022), PaLI (Chen et al., 2022), and PaLM-E (Driess et al., 2023) leverage cross-attention or soft prompts to incorporate information from other modalities. Building upon these techniques, recent work has explored augmenting LLMs with modules specifically designed for reasoning. For example, CALM (Bansal et al., 2024) employs cross-attention between specialized and general models to enhance the general model's capabilities.

### 4.5. Hypernetworks for Parameter Generation

Our work shares conceptual similarities with the concept of hypernetworks, where a separate network (the hypernetwork) generates the parameters of another network. In the context of LLM augmentation, rather than learning a fixed set of parameters for a module, a hypernetwork could generate these parameters conditioned on embeddings representing different tasks, inputs, or contexts (Ha et al., 2017; Platanios et al., 2018). This allows for a form of parameter sharing and "entanglement" between modules that are otherwise disjoint in their parameters (Goyal et al., 2021). Hypernetworks have been used to condition parameter generation on inputs as well. Examples include conditional batch normalization (de Vries et al., 2017), feature-wise

linear modulation (FiLM) for text-and-vision tasks (Perez et al., 2018), and self-modulation in GANs (Chen et al., 2019). Bertinetto et al. (2016) even conditioned parameter generation on individual examples for one-shot learning. In the context of LLMs, hypernetworks have generated diverse module parameters, such as classifier heads (Ponti et al., 2021), continuous prompts (He et al., 2022), and adapter layers (Üstün et al., 2020; Ansell et al., 2021; Mahabadi et al., 2021), conditioned on task or language embeddings. These embeddings can be learned or fixed, incorporating side information about task or language relationships.

Importantly, our approach can be viewed through the lens of hypernetworks, with the coprocessor itself acting as a hypernetwork that is conditioned on the kv-cache of the frozen LLM. Instead of generating parameters for a separate module, the coprocessor generates latent embeddings that augment the kv-cache. However, the core principle remains similar: a network dynamically generating outputs based on a rich contextual input. In our case, the kv-cache serves as a highly informative representation of the input sequence, allowing the coprocessor (hypernetwork) to generate augmentations tailored to the specific context.

## 5. Conclusion

This paper introduces differentiable cache augmentation, a novel method for enhancing frozen decoder-only language models by incorporating a learned coprocessor that operates on the model's kv-cache. This coprocessor generates latent embeddings that enrich the context provided to the LLM, improving its ability to reason and predict future tokens without requiring any modifications to the original model architecture. Our experiments demonstrate that this approach consistently reduces perplexity and significantly improves performance on a variety of reasoning-intensive tasks, even in zero/few-shot settings. These improvements are particularly notable on tasks requiring complex reasoning, such as MMLU and GSM8K.

Importantly, because the coprocessor operates offline and asynchronously, it opens up exciting possibilities for future research into models that can perform more deliberate and computationally intensive reasoning processes, including deliberation not necessarily conditioned on responding to a particular prompt. Future work will explore scaling the coprocessor to larger models or using many modular coprocessors, investigating different coprocessor architectures, and applying this method to more diverse downstream tasks. This includes investigating configurations where the coprocessor might be significantly smaller (or larger) than the base LLM, which would require exploring techniques for adapting KV-cache information between models of heterogeneous sizes, such as projection layers or sub-sampling strategies. Additionally, while our current approach gener-

ates all latent embeddings for a given augmentation point in parallel in a single forward pass for efficiency, investigating the sequential generation of these latent embeddings is an important avenue for future research. Such an approach, where latent embeddings are generated iteratively and can attend to previously generated ones within the same deliberation step, could enable more complex and nuanced internal 'thought' sequences and facilitate iterative refinement within the latent space, though it may introduce latency considerations. Finally, for efficient implementation of the structured sparse attention masks employed during training, particularly with more complex latent interactions, techniques like FlashAttention (Dao et al., 2022) variants could be explored.

## Acknowledgement

We would like to acknowledge several individuals who contributed to this work. We thank Diane Wan, Matt Barnes, and Harrison Lee for engaging in early discussions that helped shape the initial ideas. We are also sincerely grateful to Shawn Obanion, Abhinav Rastogi, Bradly Green, Jeremiah Harmsen, and Marc'aurelio Ranzato for their invaluable guidance and support throughout the completion of this work.

## Impact Statement

Improved reasoning abilities in LLMs could lead to various societal consequences, both positive and negative. On the one hand, such models could be beneficial in applications requiring complex problem-solving, such as education, scientific research, and automated decision-making. On the other hand, there are concerns about potential misuse, including generating harmful content, amplifying biases, and spreading misinformation.

Furthermore, it is important to highlight that our approach performs reasoning in latent space, meaning the reasoning trace is not readily interpretable. This lack of transparency may create additional challenges in understanding and debugging incorrect outputs generated by the LLM.

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

# A. Appendix

## A.1. Scaling with Training Data

The scaling of performance with increasing training data is a crucial aspect of evaluating the effectiveness of our approach. Figure 5 demonstrates the impact of training duration on both GSM8K accuracy and validation perplexity for our method. The x-axis represents the total number of training steps for the coprocessor. The baseline performance, representing the frozen Gemma-2 2B model, is shown for reference at corresponding intervals along this axis. As shown, we observe a clear trend of improved performance for our method with increased training data (i.e., more tokens seen during training of the coprocessor). Specifically, our method ("Ours") demonstrates a clear benefit from increased training exposure, with GSM8K accuracy exhibiting a consistent upward trend and validation perplexity showing a decreasing trend. This indicates that the coprocessor learns to generate more useful latent embeddings and better integrate with the frozen LLM as it is exposed to more data, improving next token prediction. This trend highlights the importance of scaling with training data for our approach.

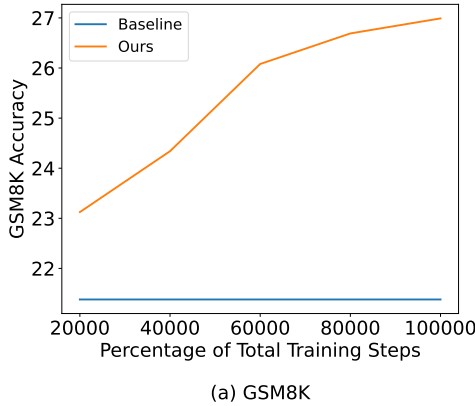 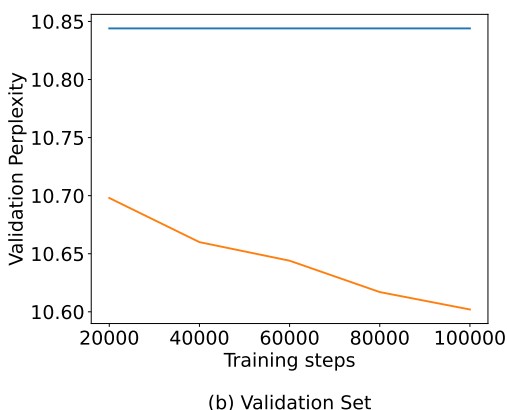

(a) GSM8K                  (b) Validation Set

*Figure 5.* Scaling of GSM8K accuracy and validation perplexity with increasing training steps for the coprocessor (using 32 latent embeddings). The baseline performance of the frozen Gemma-2 2B model is shown for reference.

## A.2. Adaptation to Downstream Tasks

All experiments described thus far have focused on training the coprocessor using the pretraining dataset. To assess the adaptability of our approach to downstream tasks, we conducted experiments using a data mixture containing the training sets of the GSM8K and MATH (Hendrycks et al., 2021) datasets. We employed LoRA finetuning (with a rank of 128) on both the baseline model and our augmented model. For the baseline, LoRA was applied directly to the base LLM, while for our augmented model, LoRA was applied specifically to the coprocessor, leaving the base LLM frozen.

Figure 6 presents the results of this downstream adaptation. We observe a substantial improvement in performance for our augmented model compared to the baseline after LoRA finetuning. This improvement is likely attributable to the strong regularization imposed by keeping the base LLM frozen during coprocessor training. This freezing prevents overfitting to the relatively small downstream datasets, allowing the coprocessor to effectively learn task-specific reasoning patterns without disrupting the general knowledge encoded in the pre-

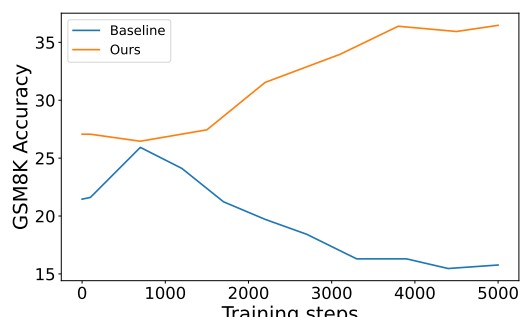

*Figure 6.* Accuracy on GSM8K's test set after LoRA finetuning. Our augmented model shows a significant improvement compared to the baseline.

trained LLM. The baseline model, with LoRA applied directly to the LLM, likely suffers from overfitting to the downstream data, limiting its performance gains. These results demonstrate the effectiveness of our approach in adapting to downstream tasks while maintaining the robustness of the pretrained LLM.

## A.3. Training Coprocessor from Scratch

We observed performance gains across most benchmarks when training the coprocessor from scratch (with randomly initialized weights), but finetuning from the pretrained LLM consistently yielded better results.

| Benchmark | Metric | Baseline | 4 Latents | 8 Latents | 16 Latents | 32 Latents | 64 Latents |
|---|---|---|---|---|---|---|---|
| MMLU | 5-shot | 52.00 | 52.03 (+0.03) | 52.21 (+0.21) | 52.75 (+0.75) | 53.55 (+1.55) | 56.63 (+4.63) |
| GSM8K | 8-shot | 21.38 | 22.52 (+1.14) | 22.59 (+1.21) | 24.41 (+3.03) | 25.78 (+4.40) | 29.80 (+8.42) |
| ARC-e | 0-shot | 80.56 | 81.69 (+1.13) | 81.86 (+1.30) | 82.79 (+2.23) | 83.12 (+2.56) | 83.21 (+2.65) |
| ARC-c | 0-shot | 50.26 | 51.71 (+1.45) | 52.22 (+1.96) | 52.47 (+2.21) | 54.27 (+4.01) | 53.24 (+2.98) |
| MATH | 4-shot | 16.50 | 16.22 (-0.28) | 16.46 (-0.04) | 16.92 (+0.42) | 17.18 (+0.68) | 18.34 (+1.84) |
| Winogrande | 0-shot | 64.01 | 65.19 (+1.18) | 65.98 (+1.97) | 66.54 (+2.53) | 66.69 (+2.68) | 67.25 (+3.24) |
| PIQA | 0-shot | 78.18 | 78.13 (-0.05) | 79.00 (+0.82) | 79.16 (+0.98) | 79.27 (+1.09) | 79.22 (+1.04) |
| SIQA | 0-shot | 51.79 | 51.94 (+0.15) | 51.64 (-0.15) | 51.84 (+0.05) | 51.94 (+0.15) | 51.89 (+0.10) |
| HellaSwag | 0-shot | 73.77 | 74.37 (+0.60) | 74.68 (+0.91) | 74.82 (+1.05) | 74.89 (+1.12) | 75.18 (+1.41) |
| Boolq | 0-shot | 75.41 | 75.66 (+0.25) | 76.94 (+1.53) | 76.97 (+1.56) | 77.80 (+2.39) | 77.46 (+2.05) |
| MBPP | 3-shot | 30.40 | 30.40 (0.00) | 30.60 (+0.20) | 30.80 (+0.40) | 32.00 (+1.60) | 32.60 (+2.20) |
| AGIEval | 3-5-shot | 31.71 | 32.52 (+0.81) | 32.22 (+0.51) | 31.92 (+0.21) | 32.78 (+1.07) | 32.35 (+0.64) |
| TriviaQA | 5-shot | 60.29 | 60.53 (+0.24) | 60.95 (+0.66) | 61.45 (+1.16) | 61.93 (+1.64) | 62.62 (+2.33) |
| NQ | 5-shot | 17.14 | 17.26 (+0.12) | 17.89 (+0.75) | 18.47 (+1.33) | 18.68 (+1.54) | 19.00 (+1.86) |
| HumanEval | pass@1 | 19.51 | 18.29 (-1.22) | 18.90 (-0.61) | 20.73 (+1.22) | 19.51 (0.00) | 19.51 (0.00) |
| BBH | 3-shot | 42.22 | 42.16 (-0.06) | 42.24 (+0.02) | 42.42 (+0.20) | 43.19 (+0.97) | 42.93 (+0.71) |

*Table 7.* Performance of baseline and augmented models across various benchmarks with coprocessor training from scratch. Check Table 2 for more detailed description.

## A.4. 9B Model Experiments

We also experimented with scaling up our approach to a larger LM, Gemma-2 9B. Due to limited compute resources, we trained the 9B model with the same batch size and training steps as the 2B model. Despite this constraint, we observed meaningful improvements on many downstream tasks. For example, on GSM8K 8-shot, the accuracy improved from 64.7% to 69.4%, and on MMLU, it increased from 71.3% to 74.5%. These results suggest that our approach can effectively enhance the reasoning capabilities of larger LMs, even with limited compute resources for training the coprocessor.

## A.5. Asynchronous Coprocessor

To assess the practicality of deploying our coprocessor in a real-world setting, we evaluated its performance when operating asynchronously alongside the base LLM. This configuration can hide the extra latency introduced from cache augmentation, as response sampling won't be blocked by the coprocessor's operation.

We simulated the case where the LLM performs N extra sampling steps (ranging from 1 to 3) before receiving the soft embedding from the asynchronous coprocessor. Table 8 presents the results on GSM8K, showing a slight degradation in accuracy as N increases.

This performance drop is likely because our training procedure did not explicitly account for this asynchronous behavior. However, it's important to note that even with this degradation, the performance remains significantly better than the baseline frozen Gemma 2B model. This experiment demonstrates the potential for hiding the coprocessor's latency through asynchronous operation, paving the way for a more efficient and practical deployment.

| | Baseline | No Async | N=1 | N=2 | N=3 |
|---|---|---|---|---|---|
| GSM8K Accuracy | 21.38 | 31.43 | 27.2 | 27.1 | 26.9 |

*Table 8.* Performance of asynchronous coprocessor on GSM8K.

