# OpenReview forum: "Deliberation in Latent Space via Differentiable Cache Augmentation"
_ICML.cc/2025/Conference — ICML 2025 poster_

### Official Review · Reviewer_vZqn · 2025-03-10

**Overall Recommendation:** 3

**Summary:**

This work proposes a novel framework to augment an existing pretrained LLM with a differentiable cache module that can be finetuned on a set of data to improve performance of the model when its combined with aforementioned cache extender. Authors claim that the novel design of the module and look-ahead training objective allows the final model to perform reasoning in the latent space introduced by the cache processor. The cache processor operate by taking in the kv cache of a given prompt (x) input, and producing a sequence of latent embeddings that are appended to the existing representations computed by the frozen LLM model.
Experiments and analysis done with proprietary data and public benchmarks show that the combined model improves the performance compared to the frozen LLM itself, as well a few other methods known from previous work.

**Claims And Evidence:**

* The idea of adding more trainable parameters to allow the model to gain extra performance improvements has clear reasoning behind it. For instance, deep seek v3 showed how multi token prediction with little parameter overhead gives significant improvements to the base model. Here we see how extra latent representation conditioned on the input is useful for predicting the output sequence.

* A big concern is lack of clarify about the "proprietary training and validation data". This weakens all experimental results as its unclear how that data is aligned with public benchmarks used in the experiments. Given that there exists lots of open source pretraining data (such as dclm), I don't see any benefit of using such data setup in a research paper aimed at sharing publicly in the conference.

* A big concern is lack of baselines showing how would a frozen model finetuned on exactly the same data perform on the benchmarks. Given that cache processor model shares the arch (more on this in the next point) with the main LLM, such baseline is required.

* Experiment descriptions do not mention much of details about the parameter capacity of cache processor w.r.t. the main model as well as competing methods such as LoRA. Its crucial to be shared as its one of the main drivers behind the model improvements: adding cache processor adds more parameters, so another baseline would be scaling up the frozen model with more params to match cache processor. While authors claim the efficiency of async cache processor abilities, its still a valid baseline that can serve as an upper bound of the performance.

**Essential References Not Discussed:**

The presented discussion provides enough context for this work.

**Experimental Designs Or Analyses:**

The choice of training and validation data make it hard for readers to understand the influence of that data on the analysis of the method. This is one of the biggest concerns I have regarding this work.

**Methods And Evaluation Criteria:**

Public benchmarks make sense and allows readers to understand the ballpark of the performance, and reproduce evaluations if needed.

**Other Comments Or Suggestions:**

none

**Other Strengths And Weaknesses:**

This method might open room for future work that'd train a number of such "expert" cache processors that can be trained separately on specific data domains to contribute towards the final mixture model.

**Questions For Authors:**

* I'd like to increase my score as long as authors would consider designing experiments using open sourced training datasets that are known to not be contaminated with any benchmark data. If thats not possible, would authors be able to confirm that they did extensive de-contamination of their data w.r.t. all benchmark tasks from table 2?

* Could you please include more information about the capacity (number of parameters) of cache processors that were trained in the experimental models?

* Could you please provide any baseline result showing the performance of the baseline/frozen LLM if it was finetuned using same data and same hyper-parameters?

**Relation To Broader Scientific Literature:**

Related method section is well written and address different alternative methods such as reasoning in discrete space with CoT, kv cache compression for model efficiency; those method address features of the proposed method from different angles.

**Theoretical Claims:**

There are no theoretical claims with corresponding proofs in the manuscript.

---

> ### Author Rebuttal · Authors · 2025-03-27
>
> Dear Reviewer vZqn,
>
> We sincerely thank you for your detailed review and insightful comments on our work (Submission 403). We appreciate you acknowledging the merit of our core idea, the quality of the related work section, and the potential for future work stemming from our method.
>
> We understand your primary concerns revolve around the proprietary dataset used and the need for specific baselines and parameter details. We appreciate the opportunity to clarify these points.
>
> ### **Regarding Proprietary Data and Fine-tuning Baseline:**
>
> We recognize your significant concerns regarding the clarity of the "proprietary training and validation data," its potential influence, and the corresponding lack of a baseline showing the frozen model fine-tuned on this same data. These points were raised multiple times and in your third question.
>
> **Our Response:**
> > We sincerely thank the reviewer for highlighting these concerns regarding our dataset and baselines. We understand that clarity here is crucial for evaluating our method's contribution.
>
> > The dataset used to train the coprocessor is *a subset of the data the model was pretrained on*. We did perform the suggested experiment of further finetuning the base model on this same dataset subset. However, as the base model was already extensively optimized on this data distribution during its initial pretraining, we observed negligible differences in perplexity or downstream performance. Standard fine-tuning paradigms predominantly yielded diminishing returns when applied directly to the original pretraining corpus. Furthermore, consistent with typical large-scale pretraining data, the corpus we used consists of text for self-supervised next-token prediction and lacks curated instruction-following examples or specific question-answering formats as in downstream tasks.
>
> > Given that this fine-tuning baseline showed no significant change from the original frozen model, we initially omitted it to maintain focus on comparisons highlighting the architectural contribution of the coprocessor. However, we understand the reviewer's point about baseline completeness. We will update the manuscript (e.g., in the dataset description or appendix) to clarify the nature of this dataset (as part of the original pretraining corpus, focused on self-supervision) and explicitly state the result of the fine-tuning baseline experiment, explaining why it yielded minimal change.
>
> ### **Regarding Parameter Counts and Scaled Baseline:**
>
> Regarding your request for more details on parameter capacity and the suggestion of a scaled-up baseline, as asked in your second question:
>
> **Our Response:**
> > We thank the reviewer for the important question on parameter counts. In our main experiments, the coprocessor has the same number of parameters as the base LLM.
> > While a baseline scaling the base model to ≈2x size is a relevant comparison point, our primary goal was specifically enhancing and augmenting an existing, frozen model, rather than the distinct and computationally intensive task of training a larger model from scratch, placing the scaled baseline outside this study's scope.
> > Importantly, we also explored parameter efficiency using a LoRA variant, which added only ≈2% of the base model's parameters while still providing substantial performance gains.
> > We will revise the paper to clearly state all relevant parameter counts (full coprocessor and LoRA variant) and briefly discuss the scaled baseline context.
>
> ### **Regarding Data Decontamination / Use of Open Data:**
>
> Addressing your first question regarding data decontamination, the potential use of open-source datasets, and the condition for increasing your score:
>
> **Our Response:**
> > We appreciate the reviewer's specific condition for potential score improvement. While re-running our main experiments on large open-source datasets is infeasible within the rebuttal period (though valuable future work), we confirm that we performed extensive decontamination of our proprietary pretraining data specifically against all benchmark tasks reported in Table 2.
>
> ---
>
> We hope these clarifications, the results from experiments already performed (such as the fine-tuning baseline), and our commitments to update the manuscript with further details (data description, parameter counts, decontamination confirmation) adequately address the main concerns raised. We thank you again for your constructive feedback and willingness to reconsider your score based on these clarifications.
>
> Sincerely,
>
> The Authors of Submission 403

---

> > ### Comment · Reviewer_vZqn · 2025-04-05
> >
> > Thanks for addressing my comments and looking forward to see the changes you plan to add. I will increase my score.

---

### Official Review · Reviewer_9maT · 2025-03-13

**Overall Recommendation:** 4

**Summary:**

This paper proposes a novel method that augment the memories (kv-caches) with a set of latent embeddings from auxiliary compressor modules. This offers two main advantages, end-to-end differentiability by using soft (continuous) tokens and asynchronous operation by using compressor in offline while freezing the base model. Without fine-tuning on downstream tasks, they showed good performance improvement across diverse tasks.

**Claims And Evidence:**

The claims and their motivations make sense a lot. Especially, using extra modules, which can be used asynchronously in potential, can efficiently improve the performance through ‘latent thinking’.

**Essential References Not Discussed:**

Like above explanations, there are some concurrent works that use extra modules (LoRA or assistant LLM) to generate thought tokens. I’d just suggest the authors to cite these references in their paper.

[1] Compressed Chain of Thought: Efficient Reasoning Through Dense Representations

[2] SoftCoT: Soft Chain-of-Thought for Efficient Reasoning with LLMs

**Experimental Designs Or Analyses:**

- Analysis on computational costs would make these methods more validate.

**Methods And Evaluation Criteria:**

Methods are well aligned to their motivations. Yet, I have some following concerns and questions:

- How about the training speed for these methods? KV-caches from frozen base models are reused in coprocessor, while coprocessor is being updated. So, it seems like we have to run additional forward step with frozen LLM for every samples. Though there could be advantages like no need to update some parameters related to KV caches, or reduced memory footprint for running coprocessor.
- The authors suggested these methods can be utilized asynchronously, but it would be good to elaborate inference process for it in Section A.5. If there are extra sampling steps in base model, which positions do we put latent embeddings?
- Have you tried to output latent embeddings sequentially, not in parallel?
- One disadvantage could be coprocessor should be the same size with base model due to reusing KV caches and augmenting soft embeddings. As this point can make these methods not to be scalable, further discussions for heterogeneous model sizes would be great.
- Is there any result that generate and use latent embeddings for multiple times during generation, not just a single forward call like in main experiments. I guess we have to adjust attention masks if we generate additional latent embeddings as latent embeddings do not attend each other in training settings. This could induce additional overhead during inference.
- What were the results when you train the models with 1 ahead token and cache augmentation techniques (regarding to Table 6 results)?

I cannot see any problems in evaluation criteria.

**Other Comments Or Suggestions:**

N/A

**Other Strengths And Weaknesses:**

N/A

**Questions For Authors:**

I’d be glad to discuss with the authors to raise my evaluation score. Please refer to some questions above.

One further question is how the authors implement parallel decoding parts efficiently. Is it possible to use FlashAttention mechanism for this non-causal attention masking?

**Relation To Broader Scientific Literature:**

I believe this work has been related to recent latent reasoning research. Especially, this work can be classified to latent reasoning methods that use extra modules to augment thought tokens. CCOT and SoftCoT (see below references) seem well aligned with this paper’s research direction.

**Theoretical Claims:**

N/A

---

> ### Author Rebuttal · Authors · 2025-03-29
>
> Dear Reviewer 9maT,
>
> Thank you for the positive evaluation, thoughtful review, and examination of the supplement. We appreciate your constructive questions and address them below:
>
> ### **Regarding Training Speed/Process:**
>
> You asked about the training speed and the forward passes involved.
>
> **Our Response:**
> > We thank the reviewer for the question regarding training speed. The training involves three main forward passes as outlined in Figures 1 and 2: (1) LLM processes input text to generate KV caches. (2) The coprocessor takes these KV caches to generate latent embeddings in parallel for all insertion points using an attention mask (avoiding extra forward steps per sample). (3) Latent embeddings are combined with the original text KV caches in the LLM to predict ahead tokens for loss calculation. Due to KV cache reuse, the effective sequence length involves the original length plus terms related to the number of latent and ahead tokens per insertion point. We will ensure this process is clearly described.
>
> ### **Regarding Asynchronous Inference Elaboration:**
>
> You asked for elaboration on the asynchronous inference process and the placement of latent embeddings when extra sampling steps occur.
>
> **Our Response:**
> > Thank you for this important question about asynchronous inference. To clarify the process: If, for example, the base model performs N=2 asynchronous sampling steps immediately after the prompt while the coprocessor computes, the coprocessor uses the KV cache of the *original prompt only* (ignoring the N async steps) to generate latent embeddings in a single pass. These generated latents are then inserted *between* the original prompt's KV cache and the KV cache of the N=2 asynchronously sampled tokens before resuming further generation. We will add a detailed elaboration of this process to Appendix A.5 as requested.
>
> ### **Regarding Sequential Latent Embedding Generation:**
>
> You asked if we tried outputting latent embeddings sequentially.
>
> **Our Response:**
> > We thank the reviewer for asking about sequential latent generation. We did not pursue this approach in the current work primarily due to the significant latency costs associated with sequential generation during both training and inference compared to our parallel method. However, we agree that exploring sequential latent deliberation is an interesting direction for future research, potentially offering different trade-offs.
>
> ### **Regarding Scalability and Heterogeneous Model Sizes:**
>
> You raised a valid point about the potential disadvantage of the coprocessor needing to be the same size as the base model and asked for discussion on heterogeneous sizes.
>
> **Our Response:**
> > Adapting to different coprocessor/base model sizes (requiring KV cache adaptation like sub-sampling) needs further research. We acknowledge this limitation and will add discussion on scalability considerations for heterogeneous sizes.
>
> ### **Regarding Multi-step Latent Generation:**
>
> You asked about generating and using latent embeddings multiple times during generation.
>
> **Our Response:**
> > Our training uses single-step parallel latent generation for efficiency, as multi-step dependencies increase training compute. Multi-step generation (where latents attend to prior latents) is compelling future work, especially for dialogue, though evaluating it requires infra changes beyond this rebuttal. We will discuss this possibility as future work.
>
> ### **Regarding results of 1-ahead + Cache Augmentation:**
>
> You asked about the results when training with 1-ahead token prediction combined with cache augmentation.
>
> **Our Response:**
> > Thank you for this specific question regarding Table 6. When training with only 1 ahead token combined with cache augmentation, we observed only negligible improvements over the baseline model.
>
> ### **Regarding Suggested References:**
>
> You suggested citing concurrent works like CCOT and SoftCoT.
>
> **Our Response:**
> > Thank you for suggesting CCOT and SoftCoT. We agree they are relevant and will cite/discuss them appropriately in the revised related work section.
>
> ### **Regarding Efficient Parallel Decoding Implementation:**
>
> You had a further question on implementing the parallel decoding efficiently, specifically regarding FlashAttention.
>
> **Our Response:**
>
> > Thank you for the question about efficient implementation. While we use standard attention optimizations, specialized techniques like FlashMask offer a path for future efficiency gains. FlashMask is suited for our method's sparse non-causal masks, potentially improving speed/reducing overhead by skipping computation for masked connections (e.g., latent-to-latent).
> ---
>
> We hope these responses adequately address your questions and comments. Thank you again for your insightful feedback and positive evaluation, which will help us improve the paper significantly.
>
> Sincerely,
>
> The Authors of Submission 403

---

> > ### Comment · Reviewer_9maT · 2025-04-04
> >
> > Thanks for the answers. I have the following comments:
> >
> > - Regarding asynchronous inference, I understand the overall process and performance (A.5) of asynchronous inference. However, I feel a slight disconnect because the emphasis on asynchronous methods throughout the paper seems quite much (in Introduction) to the rather small experiment presented in the appendix. I fully acknowledge it's a significant strength of this framework, but the experimental support appears lacking. Is there any plan for supplementary experiments about this?
> >
> > - Regarding the performance of "1-ahead + Cache Augmentation" being the same as the baseline, what could be the reason for this? I suspect it might be related to the length of the latent embedding. It seems like the width of the 16 latent embeddings is too much for the 1-ahead token. Is there any ablation study on the number of latent embeddings? (not requiring experiments at this point)
> >
> > - Regarding the references, it was simply a suggestion as the methodologies that use external modules for latent embeddings seem very relevant, but no mention each other.
> >
> > ===========
> >
> > Thanks for the clarification. All my concerns are resolved, and I raise my score accordingly.

---

> > > ### Author Response · Authors · 2025-04-04
> > >
> > > Thanks for your reply!
> > >
> > > 1.  **Regarding Asynchronous Inference:** We agree that the capacity for async operation is an important strength of this work. However, the core technical contribution that needs to be validated is that the co-processor can be efficiently fit with a pre-training objective, on (plentiful) pre-training data, using the approach depicted in figure 2, rather than on task-specific data.  This core technical contribution allows the co-processor to be large enough to do non-trivial latent reasoning.  Therefore, our main experiments focused on showing the approach can scale.  We do give the async evaluation in the appendix to show that async operation is possible with limited performance regressions, even with the co-processor not trained "async-aware".  We do agree with the reviewer that additional work, for example re-running the co-processor training to optimize in an asynchronous-aware manner, would be interesting, but doing so before the end of the rebuttal period would require more computational resources than we currently have.
> > >
> > > 2.  **Regarding 1-ahead + Cache Augmentation:** Sorry for misunderstanding your previous question – I thought you asked about 1 latent embedding, but you were asking about 1 ahead token. The results for 1 ahead and 2 ahead tokens on GSM8K are **22.90** and **23.65**, respectively, both higher than the baseline of **21.38** (results for larger numbers of ahead tokens are in Table 6). Our observation was that having a decent number of ahead tokens is important so the latent embeddings learn more than just predicting the very next token(s). However, using a very long lookahead seemed to harm prediction performance for the first few tokens immediately after the latent embeddings, which degraded sampling performance. We do have ablation studies on the number of latent embeddings in Figure 3, Table 1, and Table 2.
> > >
> > > 3.  **Regarding References:** Thanks! We agree these are good references that we should include in our related work section.
> > >
> > > We appreciate you engaging further and hope this addresses your follow-up comments.

---

### Official Review · Reviewer_w2nr · 2025-03-14

**Overall Recommendation:** 2

**Summary:**

In this work, the authors train a hyper-network (termed “coprocessor”) which takes a KV-cache from a language model mid-generation and produces a set latent embeddings which are appended to the KV-cache before producing the final answer. Critically, the coprocessor produces the latent embeddings with *a single forward pass* (unlike CoT decoding which requires decoding tokens step-by-step).

The main claims of this work are:

1. Cache augmentation enables lower perplexity on subsequent tokens.
2. When the cache is augmented the decoder achieves improved performance compared to CoT reasoning on a range of reasoning-intensive tasks.
3. Using the coprocessor is more efficient than using CoT decoding because CoT decoding is sequential and the coprocessor operates in parallel.

**Claims And Evidence:**

Are the claims made in the submission supported by clear and convincing evidence? If not, which claims are problematic and why?

1. If the claim is about the comparison with a frozen model, this claim is well-supported by evidence (see Figure 3). However, I don’t think this is a fair baseline. The coprocessor saw over 200 billion tokens of training data from a “proprietary pretraining dataset” before evaluation on that same proprietary dataset. In contrast, the baseline did not see any of that data. Without any further information on that pretraining dataset, it is not clear to me whether the reduction in perplexity is due to the proposed methodology (i.e. the coprocessor), or just the fact that we’re training on from the same distribution as the test data. To be convinced that the benefits come from the specifics of having a coprocessor, I’d like to see how perplexity improves when we simply fine-tune the base model on the same pretraining data.
2. This claim is not very well-supported by evidence. Comparisons with CoT are only made on GSM-8K, a single reasoning task. To support the claim that cache augmentation actually improves performance w.r.t. CoT I would expect evaluations on more datasets.
3. This idea is not supported by strong empirical or theoretical evidence in the paper. The authors argue in the introduction and abstract that cache augmentation has efficiency benefits over CoT because the coprocessor can be run asynchronously while the model continues to generate. However, it’s not obvious to me that this could actual yield latency or throughput improvements in practice. At small batch sizes, language model generation is typically I/O-bound (the bottleneck is loading weights from HBM to the cores, not performing the compute). Since the coprocessor has its own weights, loading those weights will contend for bandwidth with the generation. As a result, the coprocessor and generator cannot fully overlap. At high batch sizes, we are typically compute bound (the cores are “full”), so adding more parallelism won’t help significantly. To convince me that the method provides speedup, I would expect to see empirical speed benchmarks or at least a theoretical cost model. Otherwise, I think this claim should be downplayed in the abstract and introduction.

**Essential References Not Discussed:**

None that I’m aware of. The authors provide a nice review of the relevant literature.

**Ethical Review Concerns:**

Several works (e.g. https://arxiv.org/abs/2310.07923) have shown that the power of chain of thought actually comes specifically from the sequential nature of the computation. I suspect that on some tasks which require sequential computation (e.g. S5), CoT could outperform the parallel coprocessor method. Have you found any limitations or tradeoffs when it comes to the parallel nature of the coprocessor?

**Experimental Designs Or Analyses:**

Yes I did. Please see the discussion of claims and evidence above where I discuss the experimental design and analyses in the context of the paper’s claims.

**Methods And Evaluation Criteria:**

Please see discussion of claims and evidence above.

**Other Comments Or Suggestions:**

None.

**Other Strengths And Weaknesses:**

The method proposed is interesting and to my knowledge novel. However, due to some weaknesses in the evidence discussed above, it’s not clear to me how much of the benefit comes from the proposed method vs. the choice of specific training data.

**Questions For Authors:**

Several works (e.g. https://arxiv.org/abs/2310.07923) have shown that the power of chain of thought actually comes specifically from the sequential nature of the computation. I suspect that on some tasks which require sequential computation (e.g. S5), CoT could outperform the parallel coprocessor method. Have you found any limitations or tradeoffs when it comes to the parallel nature of the coprocessor?

**Relation To Broader Scientific Literature:**

In my view, the paper is most closely related to the hypernetwork literature. The authors do a great job situating the work in the context of the broader literature in the related work.

**Theoretical Claims:**

Not applicable.

---

> ### Author Rebuttal · Authors · 2025-03-27
>
> Dear Reviewer w2nr,
>
> Thank you for the detailed review of Submission 403 and the constructive feedback. We're glad you found the method novel and the related work discussion good. We address your points below:
>
> ### **Regarding Claim 1 (Perplexity Reduction & Baseline Fairness):**
>
> You questioned the baseline fairness due to training data and suggested a fine-tuning comparison.
>
> **Our Response:**
> > We thank the reviewer for their detailed feedback and insightful question regarding the training data and baseline comparisons. We recognize the importance of providing more clarity here.
>
> > The dataset used to train the coprocessor is *a subset of the data the model was pretrained on*. We did perform the suggested experiment of further finetuning the base model on this same dataset subset. However, as the base model was already extensively optimized on this data distribution during its initial pretraining, we observed negligible differences in perplexity or downstream performance. Standard fine-tuning paradigms predominantly yielded diminishing returns when applied directly to the original pretraining corpus. Furthermore, consistent with typical large-scale pretraining data, the corpus we used consists of text for self-supervised next-token prediction and lacks curated instruction-following examples or specific question-answering formats as in downstream tasks.
>
> > Given that this fine-tuning baseline showed no significant change from the original frozen model, we initially omitted it to maintain focus on comparisons highlighting the architectural contribution of the coprocessor. However, we understand the reviewer's point about baseline completeness. We will update the manuscript (e.g., in the dataset description or appendix) to clarify the nature of this dataset (as part of the original pretraining corpus, focused on self-supervision) and explicitly state the result of the fine-tuning baseline experiment, explaining why it yielded minimal change.
>
> ### **Regarding Claim 2 (Performance Improvement vs. CoT & Evaluation Scope):**
>
> You noted the limited evaluation scope (GSM-8K only) for the CoT comparison.
>
> **Our Response:**
> > We appreciate this point. Our primary goal wasn't necessarily to universally outperform CoT, but rather to investigate parallel 'latent deliberation'. We compared against zero-shot CoT ('Let’s think step by step') because it enhances reasoning at inference time without task-specific demonstrations or fine-tuning, making it a comparable low-data approach similar to our use of general pretraining data. In our experiments, this prompt yielded near-baseline results on MATH/HumanEval; we highlighted GSM-8K because zero-shot CoT provided a notable improvement there, thus offering a meaningful comparison point against our method. We will add discussion clarifying this rationale, explicitly mention the MATH/HumanEval CoT results for completeness, and refine claims accordingly in the paper.
>
> ### **Regarding Claim 3 (Efficiency Benefits vs. CoT & Hardware Considerations):**
>
> You questioned the efficiency claim due to lack of empirical evidence considering hardware limitations.
>
> **Our Response:**
> > Thank you for raising these critical efficiency points regarding I/O/compute bounds. Your concerns are valid under the assumption of shared hardware. However, our proposed architecture does not impose this constraint. The asynchronous coprocessor performs its single forward pass (akin to prefill) and can be deployed on **separate hardware** (e.g., a different accelerator or node). This fundamentally changes the efficiency analysis by decoupling its execution from the base model's subsequent token-by-token generation. This mitigates resource contention on the decoder's device and enables potential efficiency gains not possible if constrained to the same hardware.
>
> ### **Regarding your Question (Limitations/Tradeoffs of Parallel vs. Sequential Computation):**
>
> You asked about the limitations of the parallel approach vs. sequential CoT.
>
> **Our Response:**
> > Thank you for this relevant point on CoT's sequential nature and the cited work. Our work takes a first step in exploring parallel latent deliberation. We agree it's plausible that tasks heavily reliant on iterative refinement, long causal dependencies, or where intermediate steps explicitly build upon each other might benefit more from sequential CoT. Our current parallel coprocessor doesn't explicitly model such sequential dependencies within the latent injection. This represents a potential limitation compared to sequential methods, which we will acknowledge and discuss in the revised paper. Investigating sequential latent deliberation is an important avenue for future work.
>
> We hope this addresses your concerns and thank you again for the valuable feedback to improve the paper.
>
> Sincerely,
>
> The Authors of Submission 403

---

### Official Review · Reviewer_n6p1 · 2025-03-14

**Overall Recommendation:** 4

**Summary:**

In this work a co-processor is trained to get as input the generated KV-cache of a frozen model - after given an input x  and a set of soft tokens and produce a set of latent embeddings z. These embeddings are appended to the KV-cache (augmentation) and the original frozen model decodes towards output y (generation with augmented context).

The co-processor is trained by allowing it to first generate a number of latent representation and only then produce a predetermined number of output tokens for a training loss to be computed (starting from predetermined positions). Thus augmented models exhibit lower perplexity and increased performance across a broad selection of tasks, which typically increases when allowing the model to generate more latent embeddings.

This method, which allows asynchronous operations and sports end-to-end differentiability, compares favorably to closely or remotely similar schemes (Pause Token, zero-shot COT). The co-processor is typically initialized as a twin model but can also be trained from scratch or through LoRA (and the latter although incurring a performance hit is a very attractive option).

**Claims And Evidence:**

Yes

**Essential References Not Discussed:**

N/A

**Experimental Designs Or Analyses:**

Yes

**Methods And Evaluation Criteria:**

Yes

**Other Comments Or Suggestions:**

- It would be nice to include a comment on the overhead of generating more embeddings (time) or on the fact that there are two, rather than one models in use (space)  - at inference time.

- Coconut is also a recent similar approach in the sense that the continuous latent space is explored more before the materialization of tokens. The reader would expect a more detailed comparison - basically as started towards the end of page 7.

**Other Strengths And Weaknesses:**

This is a very timely contribution because decoding towards tokens could be seen as a kind of quantization, so exploring the continuous latent space more could be very advantageous towards finally producing tokens of "higher" quality.

Training seems more complex, there are more and different hyper-parameters to explore (positions, how many latents, how many tokens to generate). This will translate to training overhead.

**Questions For Authors:**

Could you position the results obtained here in math reasoning tasks and the training method used to reach those as compared to recently re-popularized RL techniques that attain impressive results for this type of tasks? Is there room for synergy? Or these views are totally orthogonal?

**Relation To Broader Scientific Literature:**

Most relevant: further explorations of latent space; Coconut and Pause Token papers are close along this dimension.

**Theoretical Claims:**

N/A

---

> ### Author Rebuttal · Authors · 2025-03-29
>
> Dear Reviewer n6p1,
>
> Thank you very much for your positive evaluation and your insightful review of our work. We greatly appreciate your accurate summary, positive feedback on our claims and experiments, and for reviewing the supplementary material thoroughly. We are encouraged that you see this as a timely contribution.
>
> We would like to address your specific comments and question:
>
> ### **Regarding Inference Overhead (Time/Space):**
>
> You suggested including comments on the overhead related to generating embeddings and using two models at inference time.
>
> **Our Response:**
> > Thank you for raising this important point about inference overhead. Regarding space, while our main setup uses a coprocessor similar in size to the base model, we also demonstrated strong results with a parameter-efficient LoRA variant adding only ~2% extra parameters (as discussed in the supplement). Regarding time, our method adds only one extra forward pass through the coprocessor during inference. Crucially, this pass can operate asynchronously, potentially hiding much of its latency, as discussed regarding deployment scenarios. Furthermore, KV cache reuse limits the computational overhead of this pass. We will add a note discussing these inference time and space considerations more explicitly in the revised manuscript.
>
> ### **Regarding Comparison with Coconut:**
>
> You suggested a more detailed comparison with the Coconut paper.
>
> **Our Response:**
> > Thank you for the suggestion to expand the comparison with Coconut. While both works explore continuous representations, key differences make direct comparison challenging. Coconut focuses on training an LLM to generate sequential 'continuous CoT' embeddings, often fine-tuning on specific downstream tasks. In contrast, our method uses a separate coprocessor to generate *parallel* latent embeddings that augment a *frozen* base LLM, and our coprocessor is trained on general pretraining data, aiming for broad applicability without task-specific tuning. Given these methodological distinctions and the near-simultaneous appearance of the works (Dec 2024), a direct empirical comparison is difficult. However, we will expand the discussion in the related work section to better highlight these conceptual similarities and differences as requested.
>
> ### **Regarding Relation to RL Techniques for Reasoning:**
>
> You asked how our results/method for math reasoning compare to recent RL techniques and if there's potential synergy.
>
> **Our Response:**
> > Thank you for this insightful question regarding RL techniques for reasoning. Recent RL methods, particularly those using process supervision or outcome rewards, have shown impressive results, often by fine-tuning models to generate better reasoning steps or verify solutions. Our approach is largely orthogonal, focusing on enhancing the base model's capabilities through parallel 'latent deliberation' via the coprocessor, trained on general pretraining data without task-specific reward signals. However, we see potential for synergy: RL techniques could potentially be applied *on top* of our augmented model to further refine the final reasoning output or even guide the generation process. Alternatively, RL could be explored as a method to train the coprocessor itself, though this adds significant complexity. We view these as complementary approaches rather than competing ones.
>
> ---
>
> Thank you again for your valuable feedback and positive assessment of our work. We believe addressing these points will further strengthen the paper.
>
> Sincerely,
>
> The Authors of Submission 403

---

### Decision · Program_Chairs · 2025-05-01

**Decision:**

Accept (poster)

**Comment:**

This paper proposes a novel and efficient method for augmenting LLMs using a differentiable coprocessor that injects latent embeddings into the KV-cache, enabling improved reasoning without modifying the base model. Despite some concerns about baseline comparisons and proprietary data, the reviewers converged on the view that the method is timely, well-motivated, and shows strong empirical promise across multiple tasks.